# Virtual Surgical Planning and the "In-House" Rapid Prototyping Technique in Maxillofacial Surgery: The Current Situation and Future Perspectives

Fabio Maglitto [1], Giovanni Dell'Aversana Orabona [1], Umberto Committeri [1], Giovanni Salzano [1,*], Gianluca Renato De Fazio [1], Luigi Angelo Vaira [2], Vincenzo Abbate [1], Paola Bonavolontà [1], Pasquale Piombino [1] and Luigi Califano [3]

1 Department of Neurosciences, Reproductive and Odontostomatological Sciences, Federico II University of Naples, 80131 Naples, Italy; fmaglitto@gmail.com (F.M.); gdorabon@yahoo.it (G.D.O.); umbertocommitteri@gmail.com (U.C.); gianludefazio@gmail.com (G.R.D.F.); abbate.maxfacc@gmail.com (V.A.); paolabonavolonta@gmail.com (P.B.); piombino@unina.it (P.P.)
2 Maxillofacial Surgery Operative Unit, University Hospital of Sassari, 07100 Sassari, Italy; luigi.vaira@gmail.com
3 Full Professor and Head of the Department of Neurosciences, Reproductive and Odontostomatological Sciences, Federico II University of Naples, 80131 Naples, Italy; luigi.califano@unina.it
* Correspondence: giovannisalzanomd@gmail.com

**Abstract:** Background: The first applications of computer-aided design/computer-aided manufacturing (CAD/CAM) in maxillofacial surgery date back to the 1980s. Since then, virtual surgical planning (VSP) has undergone significant development and is now routinely used in daily practice. Indeed, in an extraordinary period, such as that of the current COVID-19 pandemic, it offers a valuable tool in relation to the protection of healthcare workers. In this paper we provide a comprehensive summary of the clinical applications reported in the literature and review our experience using an in-house rapid prototyping technique in the field of maxillofacial surgery. methods: Our research was focused on reconstructive surgery, traumatology (especially in relation to orbital floor and zygomatic arch fractures), and COVID-19 masks. The first step was a radiographic study. Next, computed tomography (CT) scans were segmented in order to obtain a three-dimensional (3D) model. Finally, in the editing phase, through the use of specific software, a customized device for each patient was designed and printed. results: Four reconstructive procedures were performed with a perfect fitting of the surgical device produced by means of VSP. In nine orbital floor fracture cases a good overlapping of the mesh on the orbital floor was obtained. In sixteen zygomatic arch cases the post-operative CT scan showed an excellent fitting of the device and a correct fracture reduction. Regarding the COVID-19 period, six masks and shields produced proved to provide effective protection. conclusions: The timescale and costs required for the production of our "home-made" virtual design are low, which makes this method applicable to a large number of cases, for both ordinary and extraordinary activities.

**Keywords:** 3D-printing; CAD/CAM; virtual surgical planning; maxillofacial surgery





## 1. Introduction

The complexity of facial anatomy in the three-dimensional plane has always represented a challenge for maxillofacial surgeons.

An important step forward was made in the 1970s with the development by the aerospace and automotive industries of computer-aided design/computer-aided manufacturing (CAD/CAM) technology in order to obtain greater standardization and predictability in the results [1]. At the same time, in the medical field, computed tomography (CT) captured human images in three dimensions (3D) for the first time [2]. Computer-assisted surgery has represented a significant change in daily practice, allowing the management and simulation of surgical activities in a 3D model of the facial skeleton. Its application in

the field of cranio-maxillofacial surgery can be traced back to the 1980s [3]. A more useful approach was provided by Rapid Prototyping (RP) technology, a technique used to quickly fabricate a scale model of a physical part or assembly using 3D CAD data [4]. The printing phase is usually performed by means of 3D printing or "additive layer manufacturing" technology. Using 3D volumetric data from CT or magnetic resonance (MR) imaging with a 3D scanner represents an important contribution to clinical practice. Virtual surgical planning (VSP) includes the following steps: data acquisition, CT image analysis, 3D anthropometric analysis, surgical simulation, implant/model design through CAD software, and model printing through rapid prototyping. Such software (e.g., Romexis® CMF Surgery Planmeca, Asentajankatu 6, Helsinki, Finland, ProPlan CMF®, Materialise, Technologielaan 15, Leuven, Belgium and Dolphin Imaging 3D Surgery®Infolab, Via Carmelitani Scalzi 20; Verona, Italy) is capable of simulating surgical processes, such as skeletal structure osteotomies, post-oncological reconstruction, occlusion evaluation, 3D photographic mapping, and even the response of soft tissues to skeletal reconstruction. Additionally, in an extraordinary period, such as that of the current COVID-19 pandemic, CAD/CAM technology can come to the aid of the maxillofacial surgeon, offering an easy-to-apply tool for the production of personal protective equipment for healthcare workers.

Computer-assisted surgery involves seeking a standardization of the procedure, through tools that act as a guide for the surgeon during the different steps of surgery, so avoiding the operator-dependent variable. The product obtained from 3D printing is not, in fact, a definitive device and therefore implantable, but rather an instrument necessary only for the surgery. At the same time, this device allows a customization of the surgical procedure for each patient. The result is not only the obtainment of a repeatable (standardized) procedure but also one that is customized to the individual patient.

The aim of this present study is to review our experience in maxillofacial surgery using a low-cost "home-made" rapid prototyping technique.

## 2. Materials and Methods

Our research was conducted based on an analysis of different surgical areas (Table 1, Figure 1):

A.    Reconstructive surgery
B.    Traumatology:
    B.1:    Fractures of the orbital floor
    B.2:    Fractures of the zygomatic arch
C.    The COVID-19 pandemic

**Table 1.** Our cases ordered according to pathology.

| Our experience in CAD/CAM applications in Maxillofacial Surgery | | |
| --- | --- | --- |
| **Ordinary activities** | | |
| **Application field** | **Specific application** | **Cases** |
| Reconstructive surgery | Cutting-guide | 4 |
| Traumatology | Orbital mesh conformer | 9 |
| | Zygomatic shield | 16 |
| **Extraordinary activities** | | |
| **Application field** | **Specific application** | **Cases** |
| Covid-19 pandemic | Ffp2-n95 face mask | 6 |
| | Covid-19 face shield | 6 |

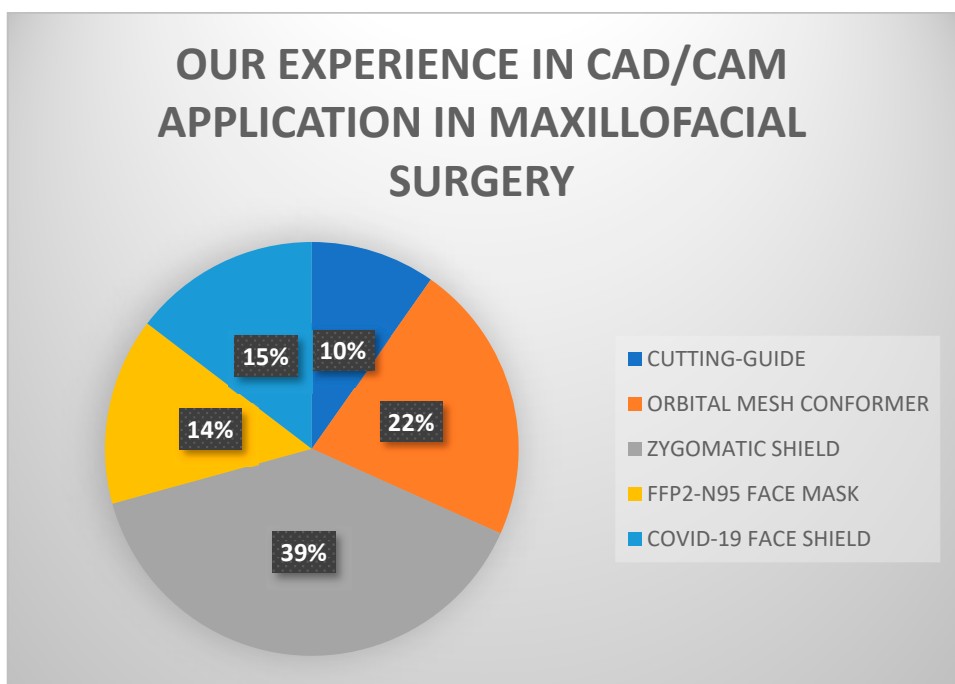

**Figure 1.** Graphical representation of our cases.

The first step was based on the study of radiographic investigations. The computer tomography (CT) scans necessary in our protocol were generated by means of high-resolution computer tomography. The CT scanner used was a Canon (ex Toshiba) 16 slice, which acquired the data with a 0.5 mm nominal slice width, reconstructed to a $1024 \times 1024$ matrix. A segmentation process based on threshold, region growing, and watershed techniques was performed on the Digital Imaging and Communications in Medicine (DICOM) files of the CT scans. This method allowed us to obtain a 3D reconstruction of the surface of interest. Our investigation involved the use of the Invesalius software, a program which involves intuitive and rapid segmentation, selecting the appropriate window (skin rather than bone) [4].

CT DICOM files were used. Once the 3D model had been obtained, it was exported as a Standard Triangulation Language (STL) file. In the editing phase, the Meshmixer software proved to be extremely useful, as well as easy to apply [5]. Through specific tools such as "Extrude" and "Export", it was possible to mold and edit the shape of the device, obtaining a customized tool for each patient. This was exported and made printable in 3D using the CURA software [6] (Figure 2).

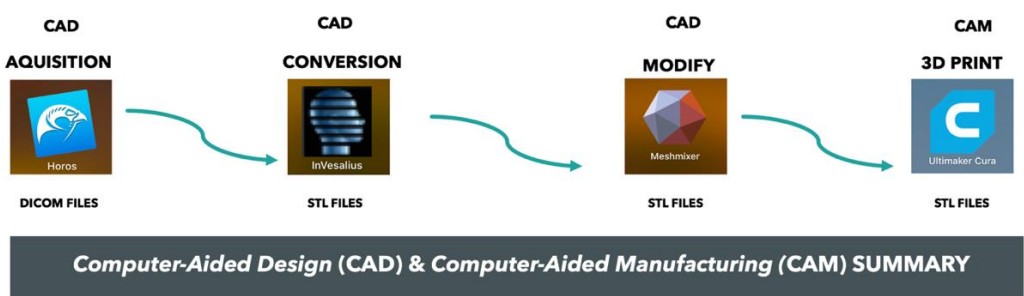

**Figure 2.** Digital workflow of our study showing in detail every application necessary for the 3D print process (every application is completely open source).

A. Reconstructive surgery in oncological cases

Between January and September 2016, four patients undergoing mandibular reconstruction with a free osteocutaneous fibula flap were enrolled in our protocol. All the patients were affected by squamous cell carcinoma of the mandibular alveolar ridge. They had been briefed on the procedure and had given their consent for the processing and registration of data in our unit of clinical research. Traditionally, this surgery is based on free-hand segmental osteotomies. The free-flap harvesting represents a challenge for the surgeon, involving a steep learning curve, with the results varying according to the surgeon's experience. The main stages of the preoperative and surgical workflow were as follows (Figure 3).

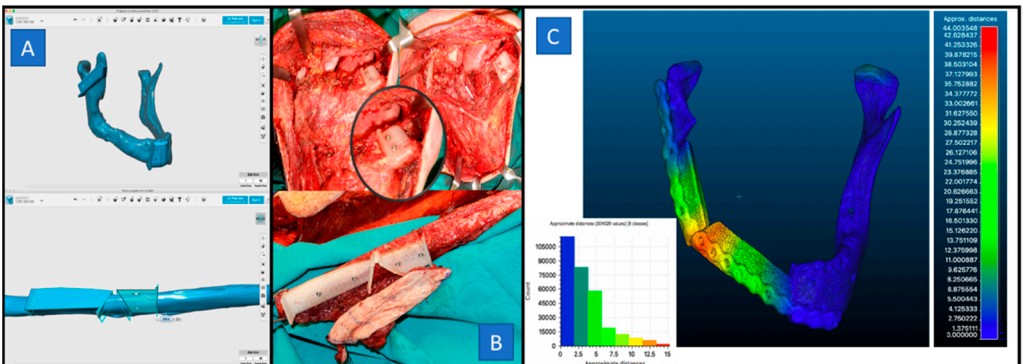

**Figure 3.** Virtual surgical planning (VSP) of the mandible and fibula cutting-guide (**A**), surgical procedure using the 3D-printed cutting-guides (**B**), and virtual comparison between the surgical planning and post-operative computer tomography (CT) to check the level of accuracy of the VSP (**C**).

(1) Transformation of the DICOM data into STL files.
(2) Design of virtual osteotomies and surgical guides. The surgical guides were molded directly on the mandibular surface in correspondence with the defined osteotomy lines under control CT recurring to the margin of oncological radicality.
(3) Production of surgical guides. Virtual cutting guides for both the jaw and fibula were exported as STL files. These STL files were imported into the CURA software and optimized for an ULTIMAKER 2 3D printer. The material used was 2.85 mm Bioflex medical PLA flexible filament (polylactic acid) (ISO 10993-5: 2009).
(4) Surgery. This involved a modified radical lymphadenectomy in all cases ipsilateral to the lesion. The mandibular osteotomy was performed along the flange of the guide using a surgical saw. The fibular osteotomy was performed after attaching the cutting guide on the fibula. The plates were molded manually following the outline of the segments of the fibula positioned to fill the surgical gap, as previously digitally programmed.

B.1 Traumatology: Orbital floor fractures

Between September 2016 and November 2018, a total of nine patients requiring surgical treatment for the reconstruction of the orbital floor were hospitalized at our department. Four of these patients met the inclusion criteria and were enrolled in the study (the other five were excluded because they had not been monitored for at least 12 months). In our four subjects the involvement of the orbital floor was due to silent sinus syndrome, an odontogenic Keratocystic tumor in the right maxillary sinus that had eroded the orbital floor, a fracture of the orbital floor, and maxillary fibrous dysplasia. Generally, the reduction of orbital floor fractures requires the use of preformed plates, which makes the process of fitting between the bone and plate extremely complicated. In addition, the anatomical complexity of this area makes the plate shaping very difficult, meaning that it is almost impossible to achieve a perfect reconstruction. All the patients underwent an

ophthalmological evaluation and a CT scan at the time of hospitalization. The following preoperative and surgical workflow was then applied (Figure 4).

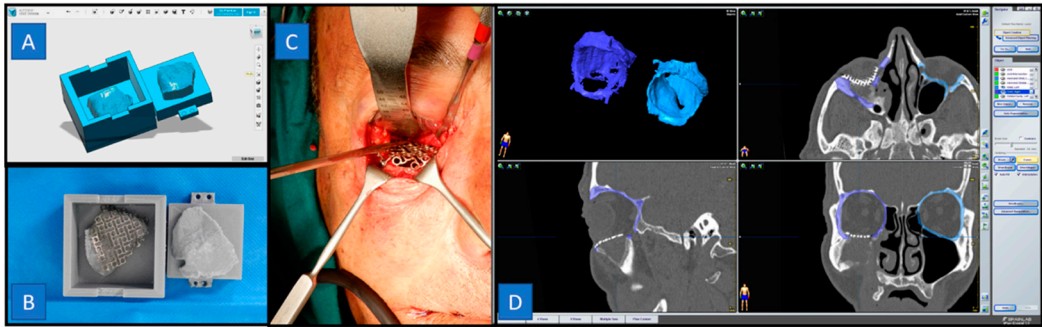

**Figure 4.** Virtual surgical planning (VSP) of the mesh-conformer (**A**), 3D printed mesh-conformer with the titanium mesh (**B**), intra-operative phase of the insertion of the pre-modeled mesh (**C**), and virtual check of the accuracy of the mesh using neuro-navigation software (**D**).

(1) Orbital segmentation and mirroring. The DICOM data of the specular contralateral orbit were used as a reference for the reconstruction of the bone defect.
(2) Processing. Through editing tools, a virtual printing machine with inlays was created to perfectly overlap the upper and lower surfaces of the eye.
(3) Slicing 3D. The virtual printing machine was exported as an STL file and optimized for the 3D jet printer extrusion with a 0.6 mm nozzle. The material used was filament 2.85 mm Bioflex medical PLA (polylactic acid) (ISO 10993-5: 2009).
(4) Mesh modeling. A 0.6 mm linear titanium mesh was modeled inside the virtual mold by applying manual pressure on the press to obtain a customized device for each patient.
(5) Surgery. All the patients underwent orbital floor reconstruction surgery under general anesthesia through a sub-ciliary incision. The mesh, previously shaped, was positioned to fill the surgical gap. The correct placement of the mesh was tested under the navigator control from the probe device.

B.2 Traumatology: Zygomatic arch fractures

Between January 2017 and February 2018 sixteen patients affected by isolated fractures of the zygomatic arch were referred to our Department. All the patients were informed about the procedure and signed a consent for treatment and data logging in our clinical research unit. All the patients underwent a preoperative CT scan as a diagnostic tool. Usually, the surgical management of isolated zygomatic arch fractures requires open-reduction treatment without fixation. To ensure adequate bone healing, a cast should be placed directly on the patient's face. The preoperative "workflow" included the following (Figure 5).

(1) Segmentation and mirroring of the skin surface.
(2) Processing. Through the "mirroring" tool, we used the contralateral zygomatic surface as a reference to develop a protective device. To ensure an adequate support for the device, we designed two slots in the upper and lower edges and two anchors in the nasal and auricular region. This volume was then exported as an STL file.
(3) Slicing. The virtual device was optimized for 3D with an extrusion jet with a 0.6 mm nozzle. The material used was 2.85 mm medical PLA bioflex filament (polylactic acid) (ISO 10993-5: 2009).

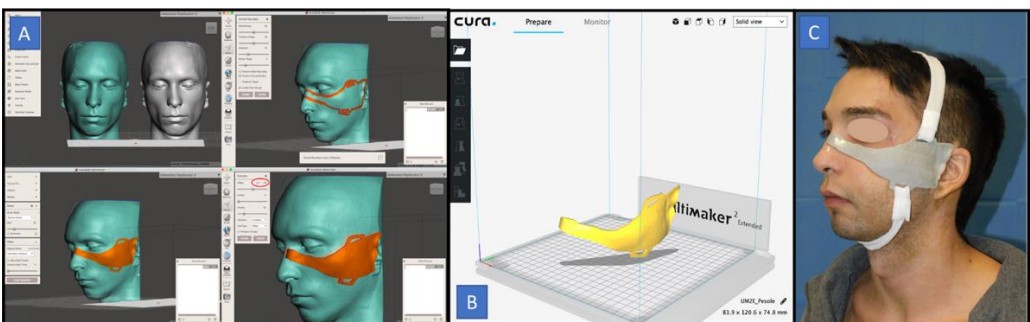

**Figure 5.** Editing phase of the virtual surgical planning (VSP) (**A**), completed shield ready for the 3D print (**B**), and face-shield perfectly fitted on the face of the patient (**C**).

C: The COVID-19 Pandemic: Face masks and face shields

Our study included six healthcare workers who underwent a 3D facial scan in order to produce a custom-made FFP2 facial mask. The entire procedure, from the design to the manufacturing of the Mask3ds, was performed at our Department. The protocol was activated during the lockdown phase of the COVID-19 pandemic from 9th March to 4th May 2020. The main phases of the protocol in detail are as follows (Figure 6).

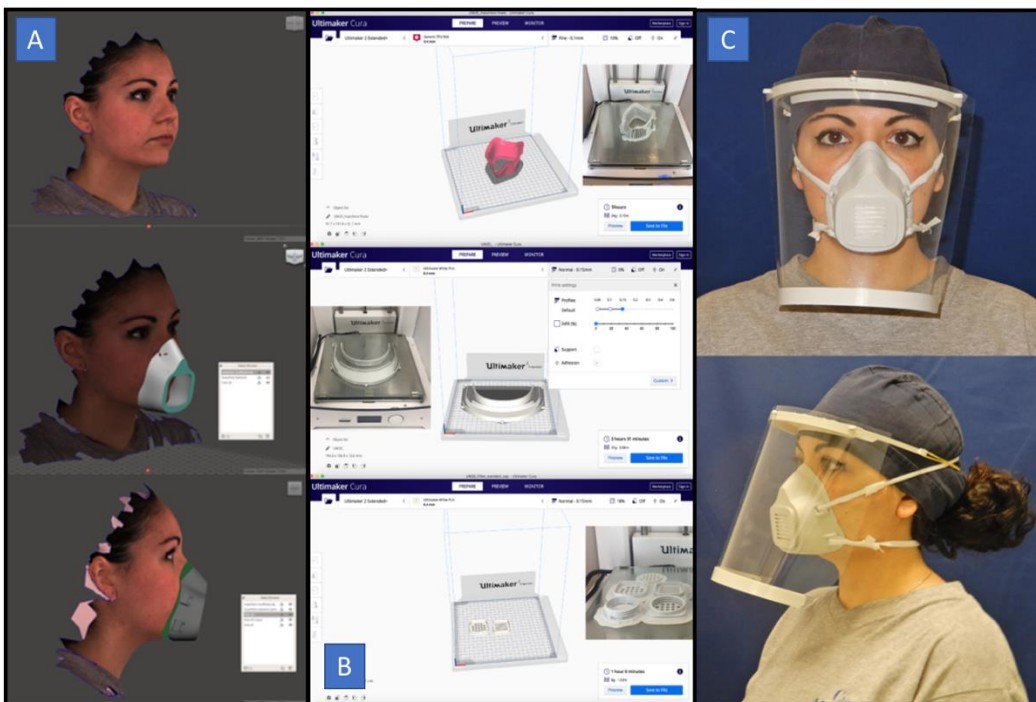

**Figure 6.** Editing phase and virtual design of the face mask (**A**), final virtual check of the COVID-19 pandemic kit (**B**), and a healthcare worker wearing the 3D-printed COVID-19 pandemic kit showing a perfect fitting of the device (**C**).

(1) Facial scanning. A facial scan was conducted using a free smartphone application (Bellus 3D, Campbell, CA, USA), obtaining an OBJ file. This application works with high-definition scanning in order to generate a ~6 MB file size (OBJ) with 60,000 triangles and a 2 K (width) color texture map. It can be used with a smartphone camera, without needing any external accessories.

(2) Editing. The OBJ file was then modified with the freeware Meshmixer Software. Using this specific tool, a customized mask for each healthcare worker was designed. Finally, this project was exported as an STL file.

(3) Slicing. The STL file was then imported into Ultimaker Cura to be printable on a 3D printer.

(4) Printing. Three elements (a mask, filter slot, and face-shield) of the Mask3d were printed using the Ultimaker 2 Extended+ 3D printer. Two types of filament were used: thermoplastic polyurethane (TPU) (Rubber TPU D27, Bioflex, Bioalfa, Soria Vecchia, Milan, Italy) and polylactic acid (PLA) (Eco PLA, 3DJake Italia, Niceshops GmbH, Paldau, Austria). A 2.85 mm TPU filament was used for the realization of the mask. The print resolution was set to 0.1 mm (fine), using a nozzle of 0.4 mm. For the filter slot and the shield, a 2.85 mm PLA filament was used, with the resolution set to 0.15 mm, using a 0.6 mm nozzle.

## 3. Results

An analysis of the results was performed based on the different surgical areas.

A Reconstructive surgery

All the patients treated with the 3D CAD/CAM printing protocol were afflicted by oral squamous cell carcinoma. Three patients were male, and one was female, with an average age of 62 years (the patients' ages ranging from 45 to 75 years). In three cases, metastatic lymphadenopathies of the neck after modified radical neck dissection (MRND) were found and post-surgical radiotherapy (RT) was performed. The mean follow-up period was 13 months. In all cases, the fibula flap was set with a skin island to reconstruct the mucous portion of the alveolar ridge. In two cases the flap of the fibula was planned with two bone segments. In the remaining cases, a "double bar" with three segments and a "single bar" with four segments were designed to restore the mandibular continuity and allow for subsequent implant prosthetic rehabilitation. All the patients underwent post-surgical CT. Using the InVesalius software, the STL files were processed to obtain the digital volume of the mandibular reconstruction. Through the Cloud-Compare software (software for the elaboration of meshes and 3D point clouds, Open-Source Project), the STL volume obtained from our digital planning procedure was superimposed on that coming from the post-surgical CT scan. Using the "alignment" tool, the highest point match was obtained and using the "mesh cloud" function the standard deviation was calculated by the software. The discrepancy between the model overlap and the points was visualized through a fusion image of the color map in all the cases treated. A standard deviation of 5.496 mm (ranging from 1.966 to 8.024 mm) was calculated from the 3D volume overlap analysis. About 6 m of Bioflex medical PLA filament were used to produce the model and the guide at a total cost of 3.6 euro per case. The total time spent designing the virtual schedule was about 3 h and the printing process and subsequent sterilization took about 6 h.

B.1 Traumatology: Orbital floor fractures

Using the iPlan Cranial 3.0 © (Brainlab, AG, USA, 2010) software, the correct position was quantified by comparing the in vivo positioned mesh with that planned digitally. A good overlap between the preoperative digital planning and the postoperative result was highlighted in all cases. The average distance between each of the three corners of the mesh and the corresponding points on the surface of the digitally programmed orbital floor was $0.71 \pm 0.23$ mm (0.566 mm, 0.533 mm, 0.766 mm, and 1 mm). Statistical analysis (ANOVA) with the Fisher's method showed a significant result at $p < 0.05$. (f-ratio value = 4.88848; *p*-value = 0.007665).

B.2 Traumatology: Fractures of the zygomatic arch

Of our group of patients, eleven (68.8%) were male and five (31.2%) were female. Analyzing the causes of the trauma, we noticed that in seven cases (43.8%) the fracture occurred after a car accident, in four cases (25%) after a sports injury, and in five cases (31.5%) after an accidental fall. The median age was 34 years (ranging from 23 to 62 years old). The post-operative CTs performed revealed an excellent positioning of the device on the skin surface and a correct reduction of the fracture in all the cases analyzed. The

tolerability of the device was evaluated with a multiple-choice questionnaire. The reliability of the questionnaire administered was confirmed by the Cronbach's alpha test with a score equal to 0.730. Only one patient left the study due to a total intolerability of the device. The average total score was 9.4.

C The COVID-19 Pandemic: Face masks and face shields

Six masks were produced and supplied to six surgeons who used them in their activities. The 3D printer took about 6 h to print the mask body, 1 h for the filter slot and 3 h for the face shield. The cost of the filament used was estimated at 1 euro for the mask body, 0.30 euro for the filter slot and 1.50 euro for the face shield. The estimated cost for the prototyping process was around 1.90 euro per case. Considering that six filters could be obtained from a single mask (0.83 euro each), adding in the price of the elastic (0.25 euro), it was possible to manufacture the final device for less than 5 euro. An FFP2 mask in Italy now costs around 5 euros. A multiple-choice questionnaire was provided to the six healthcare workers at the end of their shift. The Cronbach's alpha test was conducted in three scenarios: a medical clinic, a surgery room, and a maxillofacial surgery ward. For each item, the Cronbach's alpha coefficient was > 0.70, indicating that the mask was considered to be highly reliable.

## 4. Discussion

The scientific literature reveals that VSP is a valid and reliable method to assist surgeons in different procedures, ranging from reconstructive surgery with free flaps [5–8] to orthognathic surgery procedures [9–11], facial traumatology [12], correction of craniosynostosis [13], osteogenetic distraction [14,15], and even facial allograft transplantation [16]. In Table 2 we report the most relevant articles that have influenced and guided our research (Table 2).

The initial goal of the VSP was to produce a stereolithographic model that could assist the surgeon in reconstructing craniofacial defects or in performing orthognathic surgery [17]. In orthognathic surgery, the possibility of developing a 3D model obtained by integrating VSP represents a significant advantage thanks to the possibility of an accurate representation of facial asymmetries [18,19]. In addition, VSP provides an accurate assessment of the temporomandibular joint relations [20], which can be corrected in the event of discrepancies in the orthognathic surgery. CAD/CAM models make excellent planning possible in maxillary surgery, allowing the surgeon to guide the osteotomy and the repositioning of the jaw [21]. Mazzoni et al. used customized titanium plates and CAD/CAM cutting guides for maxillary repositioning without using splints [22]. Hanasono and Skoracki reported significant improvements in accuracy in virtually designed cases [23]. Through a comparison of the bone reference points in the post-operative phase, they detected average deviations of 4.11 mm in the simulation group and 6.92 mm in the conventional group. With the conventional method, Ayoub et al. reported, in their case history, a deviation of 6 mm. This, however, was reduced to 1.5 mm in the group with virtual design [24]. Mandibular reconstruction with free flaps represents another challenge addressed by means of this technology. The protocol we apply involves in-house prototyping in order to model the mesh based on the patient's anatomy. The time involved for the design allows a clear reduction in operating times. The accuracy of the printing depends on the accuracy of the CT, which must have slices thinner than 1–2 mm [25]. Next, the intrinsic capabilities of the printer and its components must be considered: for precision printing, a nozzle of at least 0.4 mm must be used and the accuracy of the printing profile must be close to 0.1–0.06 (fine/ultra-fine), thereby avoiding the requirement to print supports to reduce the alterations of the printing surface. Moreover, again in the area of maxillofacial trauma, we have applied this rapid in-house prototyping method to patients suffering from zygomatic fractures [26,27]. Several authors have described the use of protective masks to be applied in this area to allow adequate bone consolidation [28,29]. We therefore produced virtually, through specific software, our customized restraint device for each patient, so that it could adhere completely to the skin surface of the individual

subject. The cost of the prototyping process is around 5 euro for each case, and the time required for the CAD programming is around 90 min [25].

**Table 2.** Review of literature.

| Author | Year of Publication | Kind of Study | Population | Application Field | 3D Application |
|---|---|---|---|---|---|
| Mommaerts MY | 2001 | Retrospective study | 2 | Malformative | Planning for malformative surgery |
| Heise M | 2001 | Technical notes and review | 2 | Traumatological surgery | Protective facial shield |
| Marchetti C | 2006 | Retrospective study | 18 | Reconstructive surgery | Free fibula flap mandible reconstruction |
| Cascone P | 2008 | Technical notes and review | 1 | Traumatological surgery | Protective facial shield |
| Varol A | 2009 | Retrospective study | 5 | Osteodistraction | Planning for osteodistraction surgery |
| Edwards SP | 2010 | Systematic review | / | Imaging, reconstructive surgery, malformation surgery, orthognathic surgery, osteodistraction | Surgical planning in maxillofacial surgery |
| Antony AK | 2011 | Technical notes and review | 5 | Reconstructive surgery | Free fibula flap mandible reconstruction |
| Baker SB | 2012 | Retrospective study | 11 | Orthognathic surgery | Planning for orthognathic surgery |
| Beliakin SA | 2012 | Technical notes and review | 1 | Traumatological surgery | Virtual surgical planning |
| Doscher ME | 2013 | Retrospective study | 1 | Osteodistraction | Planning for osteodistraction surgery |
| Hsu SS | 2013 | Retrospective study | 65 | Orthognathic surgery | Planning for orthognathic surgery |
| Hanasono MM | 2013 | Retrospective study | 38 | Reconstructive surgery | Virtual surgical planning |
| Avraham T | 2014 | Retrospective review | 52 | Reconstructive surgery | Free fibula flap mandible reconstruction |
| Swennen GR | 2014 | Retrospective study | 350 | Orthognathic surgery | Planning for orthognathic surgery |
| Robdy KA | 2014 | Systematic review | / | Reconstructive surgery | Virtual surgical planning |
| Ayoub N | 2014 | Retrospective study | 20 | Reconstructive surgery | Iliac crest bone flap mandible reconstruction |
| Toto JM | 2015 | Retrospective review | 57 | Reconstructive surgery | Free fibula flap mandible reconstruction |
| Hammoudeh JA | 2015 | Systematic review | / | Orthognathic surgery | Planning for orthognathic surgery |
| Steinbacher DM | 2015 | Retrospective study | 6 | Reconstructive surgery, malformation surgery, orthognathic surgery, osteodistraction | Surgical planning in maxillofacial surgery |
| Mendez BM | 2015 | Retrospective study | 2 | Reconstructive surgery | Virtual surgical planning |
| Wilde F | 2015 | Retrospective study | 30 | Reconstructive surgery | Free fibula flap mandible reconstruction |
| Chen ST | 2015 | Retrospective study | 7 | Traumatological surgery | Virtual surgical planning |
| Chang EI | 2016 | Technical notes and review | 1 | Reconstructive surgery | Free fibula flap mandible reconstruction |
| Resnick | 2016 | Retrospective study | 43 | Orthognathic surgery | Planning for orthognathic surgery |
| Shaheen E | 2017 | Retrospective study | 20 | Orthognathic surgery | Planning for orthognathic surgery |
| Bosc R | 2017 | Retrospective study | 18 | Reconstructive surgery | Cutting guide VPS mandible reconstruction |
| Swennen GR | 2020 | Technical notes and review | 1 | COVID-19 Mask | 3D-printed mask |

VSP offers significant advantages in terms of the duration and predictability of the operation thanks to pre-formed cutting guides [30]. The benefits of CAD/CAM reconstructive procedures include the pre-surgical planning of the tumor resection and the final flap positioning. VSP proved to be more accurate and efficient than traditional surgery [31]. With this technique, similar outcomes have been obtained from surgeons with different levels of experience [32]. Indeed, the pre-surgical planning of the cutting lines and flap size allows a considerable reduction in the duration of any ischemia, related to the time previously used for the shaping of the flap [24].

The main disadvantages are the cost and delivery time of the product. In 2015, Wilde reported a cost between $2000 and $6000 for commercial products such as MIMICS (Materialize N.V., Louven, Belgium) which includes virtual planning assistance and cutting guides [33]. In 2017, Bosc reported that the complete service for the CAD/CAM system has a cost between 3000 and 5000 euro for each patient and the delivery times are longer than three weeks [34]. With such a long delivery time, the risk of cancer progression and metastasis increases.

The idea of rapid in-house prototyping can help to reduce costs and save time in performing surgery. In our study, rapid in-house prototyping and 3D-printing protocols were developed to virtually plan a mandibular reconstruction after cancer resection [25]. After the initial purchase of the 3D printer, at a cost of about 3000 euro, the cost related to the design of each case is about three euro, the sum necessary for the purchase of the material for the printing. Computer-assisted surgery also provides a greater precision in mandibular reconstruction with a clear gain for the patient in terms of functionality and quality of life [35]. However, a high level of reconstruction is difficult to achieve due to the various inaccuracies that can occur in the different phases, such as in the image acquisition, segmentation, 3D printing, surgery, and evaluation of the post-operative results [36,37].

In maxillofacial traumatology, the use of VSP has made it possible to optimize outcomes in this area with a high esthetic impact. In cases of orbital-floor fractures, a perfect reconstruction is still a challenge for the surgeon due to the possible complications that may arise, such as diplopia and enophthalmos. In the literature, several authors describe the possibility of using customized titanium meshes for orbital reconstruction [38,39]. Unfortunately, also in this case, the high cost and long prototyping times make this method inapplicable. The protocol we apply involves in-house prototyping in order to model the mesh based on the patient's anatomy. The time spent during the design allows a clear reduction in the operating time. The accuracy of the printing depends on the accuracy of the CT, which must have slices thinner than 1–2 mm.

During the health crisis related to the COVID-19 pandemic, it has been essential to protect healthcare workers by providing them with adequate personal protective equipment. Considering the main transmission pathways of SARS-CoV-2, face masks have proved to be a fundamental tool to prevent contamination [40,41]. Nowadays, there is clear evidence in the literature of the high accuracy of 3D facial scanning and it is accepted that facial digitizing procedures produce clinically acceptable outcomes for virtual treatment planning [42]. In our region, the COVID-19 emergency started on 9th March 2020 and we began to design and print the Mask3d on 16th March 2020. The perfect fitting of the mask to the skin surface allows for the breathing function to be performed exclusively through the filter unit, providing a good personal protection for the worker [43]. Using Bellus 3D, a free mobile application, the present workflow protocol is accessible to anyone. As reported by Swennen et al., the introduction of new generation smartphones with two cameras and dedicated applications, like Bellus3D, makes this system practical and available worldwide [44].

Our protocol is based on a low-cost, "home-made" rapid prototyping facial mask. The estimated cost of the prototyping process is less than 5 euro per case. Currently, due to the COVID-19 emergency, there are no certifications available. However, the use of "home-made" masks has been approved by the Italian government.

## 5. Conclusions

The aim of our analysis was not to prove the superiority of VSP, and rapid in-house prototyping, compared to the CAD/CAM methods available on the market, but, rather, to demonstrate that our workflow has led to equivalent results. The times and costs necessary for this in-house virtual design are lower, which makes this method applicable to a greater number of cases, especially when we are faced with malignant pathologies. Other advantages of 3D printing technology include the reduced surgical time durations, improved surgical outcomes, and decreased radiation exposure. The range of applications

of 3D printing continues to expand and these techniques are becoming increasingly accessible. CAD/CAM methods will drive improvements in the technology and its surgical applications, especially in the cranio-maxillofacial region.

**Author Contributions:** F.M.: head of the study project; G.D.O.: design of the study; U.C.: design of the study; G.S.: literature research; G.R.D.F.: acquisition of the data; L.A.V.: acquisition of the data; V.A.: analysis and interpretation of the data collected; P.B.: analysis and interpretation of the data collected; P.P.: drafting of the article and critical revision; L.C.: final approval and guarantor of the manuscript. All authors have read and agreed to the published version of the manuscript.

**Funding:** This research received no external funding

**Institutional Review Board Statement:** The study was conducted according to the guidelines of the Declaration of Helsinki and approved by the Institutional Review Board of Federico II University of Naples (due to the retrospective nature of this study, it was granted an exemption in writing by the University Ethics Committee. Patients' consent was obtained and can be produced on request).

**Informed Consent Statement:** An informed consent was obtained from all the subjects involved in the study. The patients in this project consented to the use of all photographs and illustrations for the purposes of educational content. The signed consent forms are on file at the Maxillofacial Department of Federico II University of Naples, Naples, Italy, containing identifying patient information and signatures. These data are available upon specific request.

**Conflicts of Interest:** The authors declare no conflict of interest.

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
