# Peer review of "Virtual Surgical Planning and the “In-House” Rapid Prototyping Technique in Maxillofacial Surgery: The Current Situation and Future Perspectives"

_applsci, doi:10.3390/app11031009_

Round 1
Reviewer 1 Report
1) Line 83: "CT scans were used to perform a segmentation process." The segmentation is performed on the scans. The sentence needs to be revised.
2) Line 84: The 3D model is not the output of segmentation but surface reconstruction. Please revise.
3) For case studies A, B.1 and B.2, please give details of the CT scan (resolution, etc.) and the specs of the X-ray scanner. For the case study C, please give the specs of the scan data (at least the resolution) and the device (smartphone with or without external accessories).
4) Lines 288-289: The print accuracy also depends on several factors other than the one mentioned in the paper. How important is the geometric accuracy of the printed part in your application? I recommend that you explain that in the paper.
Author Response
Dear Editor,
thank you for these precious corrections. We have modified our paper accordingly. In case any change is not satisfying, please do not hesitate to contact us. Here is a point-by-point response to the reviewers.
Reviewer 1 Review Report (Round 1):
We heartfully thank the reviewer for the thorough corrections. Your observations are rightful and much appreciated, giving us the opportunity to dramatically improve our work. Here is a point-by-point response to your letter:
Moderate English changes required: we provide an English revision by a native speaker moderate
- Ex Line 83 (now Line 87), We have modified this passage accordingly to your suggestion:
The first step was based on the study of radiographic investigations. The CT scans necessary in our protocol were generated by means of high-resolution computer tomography (CT).
- Ex Line 84 (now Line 93), We have modified these passages according to your observation:
This method allowed us to obtain a 3D reconstruction of the surface of interest.
- We have clarified the confusing passages according to the reviewer suggestion.
We provide to add in line ex 83 (now Line 89), (For case studies A, B.1 and B.2, please give details of the CT scan):
The CT-Scanner used was a Canon (Ex Toshiba) 16 slices, which acquired the data with a 0.5mm nominal slice width, reconstructed to a 1024x1024 matrix
Added in ex line 199(now Line 207), (For the case study C, please give the specs of the scan data):
This application works with high-definition scanning in order to generate a ~6MB file size (OBJ) with 60,000 triangles and a 2K (width) color texture map. It can be used with a smartphone camera, without needing any external accessories.
- Ex Lines 288-289 (now Line 312), We have modified this passage accordingly to your suggestion: The print accuracy also depends on several factors other than the one mentioned in the paper.
Next, the intrinsic capabilities of the printer and its components must be considered: for precision printing, a nozzle of at least 0.4mm must be used and the accuracy of the printing profile must be close to 0.1-0.06 (fine / ultra-fine), thereby avoiding the requirement to print supports to reduce the alterations of the printing surface.
We hope our changes are satisfying, otherwise we will proceed with the additional modifications you shall require. On behalf of all the authors,
Kind regards
Giovanni Salzano

Reviewer 2 Report
The authors present a meaningful workflow for in house rapid prototyping of surgical tools such as guides and molds, as well as safety equipment, using low cost open source tools.
The authors provide a detailed description of their own experience as well as a review of the literature surrounding these types of cases.
They demonstrate the way the tools were used and describe the quality of results obtained.
Overall this is a useful work to help the research community understand the potential of these tools for surgical aides.
I am generally enthusiastic about this work, however, in it's current form, it may be difficult to interpret for much of this journal's audience, who are not medically trained and do not have a clear understanding of what these procedures are, how they are typically performed and the benefit that comes from using patient specific tools. Some additional comments are below.
The quality of writing needs to be improved somewhat. There are multiple places where tense and subject verb agreement are issues.
It is an important distinction that the type of rapid prototyping tools presented here are used to make tools to assist with procedures, not implants themselves, which present a much higher standard for testing and biocompatibility. The demonstrations here include cutting guides, patient specific molds and safety tools. This distinction should be mentioned explicitly in the introduction.
Because this is not a medical journal, it would help readers understand the use cases if each surgical section described in a couple of sentences in plain language the overall purpose of each procedure and what is done when rapid prototyping is not performed. For example, what types of cutting guides are used typically, what is the drawback. How are meshes processed for orbital floor procedures typically.
28 – plan or plane?
57 – the software has not been defined and there are no citations to describe what exactly has these capabilities.
89- the authors describe the ease of use for mesh mixer but not the type of modifications they are performing using it. Please be specific. Are you changing the number of triangles to make the features exportable, are actually trimming and/or editing the shape/design?
126 – the description of this procedure is confusing. I think the authors are using the modeled when they mean molded. It is important
218 – is it possible to be more descriptive (and ideally quantitative) when describing a ‘perfect fitting’. What is typically imperfect about the fit? How much better is it here, to what precision?
Author Response
Dear Editor,
thank you for these precious corrections. We have modified our paper accordingly. In case any change is not satisfying, please do not hesitate to contact us. Here is a point-by-point response to the reviewers.
Reviewer 2 Review Report (Round 1):
Dear reviewer,
We are happy you are satisfied with our paper. Thank you for your appreciation words and for your rightful observation. We apologize for the formal mistakes we made, and we have performed the changes you required. Here is a point-by-point response to your letter:
Moderate English changes required: we provide a English revision by a native speaker moderate
We improve introduction section from ex line 63 (now Line 65):
Computer-assisted surgery involves seeking a standardization of the procedure, through tools that act as a guide for the surgeon during the different steps of surgery, so avoiding the operator-dependent variable. The product obtained from 3D printing is not, in fact, a definitive device and therefore implantable, but rather an instrument necessary only for the surgery. At the same time, this device allows a customization of the surgical procedure for each patient. The result is not only the obtainment of a repeatable (standardized) procedure but also one that is customized to the individual patient.
We have modified the M&M section according to your corrections, in order to clarify of each procedure and what is done when rapid prototyping is not performed:
Ex Line 104 (now Line 113), We have modified the passage accordingly: Traditionally, this surgery is based on free-hand segmental osteotomies. The free-flap harvesting represents a challenge for the surgeon, involving a long learning curve, with the results varying according to the surgeon’s
Ex Line 135 (now Line 145), We have modified the passage accordingly: Generally, the reduction of orbital floor fractures requires the use of preformed plates, which makes the process of fitting between the bone and plate extremely complicated. In addition, the anatomical complexity of this area makes the plate shaping very difficult, meaning that it is almost impossible to achieve a perfect reconstruction.
Ex Line 165 (now Line 178), We have modified the passage accordingly: Usually, the surgical management of isolated zygomatic arch fractures requires open-reduction treatment without fixation. To ensure an adequate bone- healing, a cast should be placed directly on the patient's face.
Ex Line 28 (now Line 27), We have modified this passage accordingly to your suggestion
Four reconstructive procedures have been performed with a perfect fitting of the surgical device produced by means of VSP.
- Line 57, We have modified these passages according to your observation: the software has not been defined and there are no citations to describe what exactly has these capabilities.
Such software (e.g., Romexis® CMF Surgery, ProPlan CMF® and Dolphin Imaging 3D Surgery®)
- Ex Line 89 (now Line 98), The observation of the author is legitimate, so we have modified these passages: the authors describe the ease of use for mesh mixer but not the type of modifications they are performing using it. Please be specific.
In the editing phase, the Meshmixer software proved to be extremely useful, as well as easy to apply [5]. (NB This sentence is repeated twice). Through specific tools such as ‘‘Extrude’’ and ‘‘Export’’, it was possible to mold and edit the shape of the device, obtaining a customized tool for each patient.
- Ex Line 126 (now Lines 124 and 134), We have modified these passages according to your observation: the description of this procedure is confusing. I think the authors are using the modeled when they mean molded.
124 The surgical guides were molded directly on the mandibular surface in correspondence with the defined osteotomy lines under control CT recurring to the margin of oncological radicality.
134 The plates were molded manually following the outline of the segments of the fibula positioned to fill the surgical gap, as previously digitally programmed.
- Ex Line 218 (now Line), We have clarified the confusing passages according to the reviewer suggestion: is it possible to be more descriptive (and ideally quantitative) when describing a ‘perfect fitting’.
All the patients underwent post-surgical CT. Using the InVesalius software, the STL files were processed to obtain the digital volume of the mandibular reconstruction. Through the “Cloud-Compare software” (software for the elaboration of meshes and 3D point clouds, Open-Source Project), the STL volume obtained from our digital planning procedure was superimposed on that coming from the post-surgical CT scan. Using the "alignment" tool, the highest point match was obtained and using the "mesh cloud" function the standard deviation was calculated by the software. The discrepancy between the model overlap and the points was visualized through a fusion image of the color map in all the cases treated.
We hope our changes are satisfying, otherwise we will proceed with the additional modifications you shall require. On behalf of all the authors,
Kind regards
Giovanni Salzano
